# Antileishmanial activity of synthetic analogs of the naturally occurring quinolone alkaloid *N*-methyl-8-methoxyflindersin

**Elaine Torres Suarez**[1�], **Diana Susana Granados-Falla**[1,2‡], **Sara María Robledo**[3�he¤a], **Javier Murillo**[3‡], **Yulieth Upegui**[3‡], **Gabriela Delgado**[1�¤b]*

1 Grupo de Investigación en Inmunotoxicología, Departamento de Farmacia, Universidad Nacional de Colombia, Bogotá, Colombia, 2 Vicerrectoría de Investigaciones, Universidad El Bosque, Bogotá, Colombia, 3 PECET, Facultad de Medicina, Universidad de Antioquia, Medellín, Colombia

he These authors contributed equally to this work.
¤a Current address: Programa de Estudio y Control de Enfermedades Tropicales, Sede Investigación Universitaria, Universidad de Antioquia, Medellín, Colombia
¤b Current address: Departamento de Farmacia, Facultad de Ciencias, Universidad Nacional de Colombia, Bogotá, Colombia
‡ These authors also contributed equally to this work.
* lgdelgadom@unal.edu.co

**Data Availability Statement:** All relevant data are within the manuscript.

## Abstract

Leishmaniasis is a neglected, parasitic tropical disease caused by an intracellular protozoan from the genus *Leishmania*. Quinoline alkaloids, secondary metabolites found in plants from the Rutaceae family, have antiparasitic activity against *Leishmania* sp. *N*-methyl-8-methoxyflindersin (**1**), isolated from the leaves of *Raputia heptaphylla* and also known as 7-methoxy-2,2-dimethyl-2H,5H,6H-pyran[3,2-c]quinolin-5-one, shows antiparasitic activity against *Leishmania* promastigotes and amastigotes. This study used *in silico* tools to identify synthetic quinoline alkaloids having structure similar to that of compound **1** and then tested these quinoline alkaloids for their *in vitro* antiparasitic activity against *Leishmania (Viannia) panamensis*, *in vivo* therapeutic response in hamsters suffering from experimental cutaneous leishmaniasis (CL), and *ex vivo* immunomodulatory potential in healthy donors' human peripheral blood (monocyte)-derived macrophages (hMDMs). Compounds **1** (natural), **2** (synthetic), and **8** (synthetic) were effective against intracellular promastigotes (9.9, 3.4, and 1.6 µg/mL medial effective concentration [$EC_{50}$], respectively) and amastigotes (5.07, 7.94, and 1.91 µg/mL $EC_{50}$, respectively). Compound **1** increased nitric oxide production in infected hMDMs and triggered necrosis-related ultrastructural alterations in intracellular amastigotes, while compound **2** stimulated oxidative breakdown in hMDMs and caused ultrastructural alterations in the parasite 4 h posttreatment, and compound **8** failed to induce macrophage modulation but selectively induced apoptosis of infected hMDMs and alterations in the intracellular parasite ultrastructure. In addition, synthetic compounds **2** and **8** improved the health of hamsters suffering from experimental CL, without evidence of treatment-associated adverse toxic effects. Therefore, synthetic compounds **2** and **8** are potential therapeutic candidates for topical treatment of CL.

**Funding:** This study was financed by Colombian Ministry of Science Technology and Innovation, E. T: Colciencias-Colfuturo grant 727-2015 for national PhD students and Colombian Ministry of Science and Technology (MinCiencias) grant 777-2017 (project 110177758192). The funders had no role in study design, data collection and analysis, decision to publish, or preparation of the manuscript.

**Competing interests:** The authors have declared that no competing interests exist.

## Introduction

Leishmaniasis is a neglected, parasitic tropical disease caused by an intracellular protozoan from the genus *Leishmania*. More than 300 million people are at risk of infection, with at least 1.3 million cases documented annually, 90% of them being related to cutaneous leishmaniasis (CL) [1, 2]. CL is characterized by one or more macular-like lesions, secondary to the bite of the phlebotomine vector, which evolve into dermal granulomas, such as papules, nodules, plaques, or skin ulcers [3, 4].

The first therapeutic choice for CL patients is based on the intravenous or intralesional administration of pentavalent antimony [5–7]. Oral administration of miltefosine or intravenous administration of amphotericin B is a second therapeutic possibility [5, 6]. However, these drugs have adverse effects, such as cardiotoxicity, hepatic damage, nephrotoxicity, or even teratogenicity (after miltefosine administration) [7, 8]. In addition, prolonged treatment schemes, parenteral administration, noncompliance and abandonment of prescribed treatment lead to the emergence of drug-resistant parasites [6, 7, 9]. Therefore, there is a need for new therapeutic alternatives that are more effective and efficient in terms of parasite elimination and disease resolution and that are safer for patients in terms of better adherence and fewer toxic effects [10, 11].

Using natural molecules and their synthetic derivatives is the main strategy followed in the search for new therapeutic options [7]. Of these, quinoline alkaloids, which are secondary metabolites found mainly in plants from the Rutaceae family, are biosynthesized from anthranilic acid and comprise carboxyl groups of anthranilic acid with an acetate group (malonate) and their subsequent cyclization of the quinolinic ring [12, 13]. Quinoline alkaloids are effective against CL caused by *L. amazonensis* and *L. venezuelensis* in BALB/c mice [14]. In addition, 2-substituted quinoline alkaloids chimanine D and B isolated from the *Galipea longiflora* K. Krause stem bark act against *L. braziliensis* and *L. donovani* promastigotes [15, 16].

The quinoline alkaloid *N*-methyl-8-methoxyflindersine (**1**), also known as 7-methoxy-2,2-dimethyl-2H,5H,6H73 pyrano[3,2-c]quinolin-5-one, is isolated from the leaves of *Raputia heptaphylla* [17]. Compound **1** has a direct effect on *L. (V.) panamensis* promastigotes and reduces the number of parasites internalized in dendritic cells [18]. However, it cannot be synthesized because of its relatively low amount extracted from its natural source [19] and its complex structure [12, 13], making it difficult to obtain sufficient amount of material for preclinical studies to validate its therapeutic potential.

This study used *in silico* tools to identify synthetic quinoline alkaloids having a structure to similar that of compound **1**. The selected compounds were them tested for their *in vitro* antiparasitic activity against *L. (V.) panamensis*, *in vivo* therapeutic response in hamsters suffering from experimental CL, and *ex vivo* immunomodulatory potential in healthy donors' human peripheral blood (monocyte)-derived macrophages (hMDMs).

## Materials and methods

### *In silico* studies

**Screening for synthetic analogs of compound 1.** We used compound **1** as the structural template. Each molecule's SMILES codes were disposed in the chemical databases ChemSpider, PubChem, and Zinc Database, and their structures were manually compared to the template. We used the Tanimoto index (TI) method to select candidate compounds by comparing the structures' dimensions and the proximity between bits in two dimensions [20]. We selected the following commercially available synthetic compounds with TI of >0.6: 1,2,3,4-tetrahydro (benzo)-3-quinolin-ol (CAS: 5423-67-6) (**2**), carboxylic acid 2-ethyl-3-propyl-4-quinolinine

(CAS: 74960-58-0) (**3**), 4-methyl-2-(1H)-quinolinone (CAS: 84909-43-3) (**4**), 4,7,8-trimetoxi-fure [2,3-b]-quinoline (CAS: 5255-76-5) (**5**), carboxylic acid-7,8-tetrahydro-quinoline (CAS: 895966-42-4) (**6**), 3,4-dimethyl-3H-imidazol [4.5f]-quinoline-2-amine (CAS: 77094-11-2) (**7**), and 2-amino-8-hydroxyquinoline (CAS: 70125-16-5) (**8**).

All compounds were purchased from Sigma-Aldrich (St. Louis, MO, USA) and Santa Cruz Biotechnology (CA, USA).

**Classification and structural analysis of quinoline alkaloid–like compounds.** We used the ChemMine Tools *(*http://chemmine.ucr.edu/tools/Clustering/*)* online service to analyze and cluster the selected compounds. Their structural descriptions were based on a group classification, considering their central structures and their most relevant substituents.

**Physicochemical properties of quinoline alkaloid–like compounds.** We used the Chem-Spider database (http://www.chemspider.com) to analyze the partition coefficient, pH-dependent partition coefficient (Log*D*), molecular weight (g/mol), acid dissociation constant (pKa), polar surface area (PSA), and solubility (mg/mL) in order to predict the biological system molecule behavior [21, 22].

## *In vitro* antileishmanial activity

**Parasites.** We kept *L. (V.) panamensis* (MHOM/COL/87/UA-140) in Roswell Park Memorial Institute (RPMI) 1640 medium (Invitrogen, Carlsbad, CA, USA) enriched with 10% fetal bovine serum (FBS; HyClone, IL, USA), 1X glutamine, and 100 U/100 μg/mL of 1% penicillin-streptomycin (Lonza, MD, USA) at 26˚C.

**Isolating hMDMs.** We isolated hMDMs from buffy coats donated by the Institute of Local Science (Biotechnology and Health Innovation, Bogotá, Colombia) and spun them in Lymphoprep density gradient medium (Stem-Cell, Vancouver, Canada). Next, we cultured mononuclear cells for 4 h in 96-well plates in RPMI 1640 medium and enriched them with 1% FBS at a density of $1 \times 10^6$ cells/mL. Subsequently, the supernatant was removed, and the cells were cultivated for 72 h at 37˚C in RPMI 1640 medium and enriched with 10% FBS, 1X glutamine, and 100 U/100 μg/mL of 1% penicillin-streptomycin at 36˚C.

**Cytotoxicity evaluation of synthetic and natural quinoline alkaloid–like compounds in hMDMs.** We cultured hMDMs in 96-well plates at a density of $5 \times 10^4$ cells/mL in 100 μL RPMI 1640 medium supplemented with 10% FBS and 100 U/100 μg/mL of 1% penicillin-streptomycin at 37˚C in a 5% $CO_2$ atmosphere. Next, we added 100 μL of this culture to each quinoline alkaloid–like compound at a concentration of 200, 20, 2, 0.2, and 0.02 μg/mL. We evaluated cytotoxicity after 72 h by adding 44 μM resazurin (Sigma-Aldrich) and analyzing the reduction to resorufin (in viable cells) using a 588 nm read on a spectrofluorometer (Tecan Genius, Tecan, Switzerland). Cells treated with amphotericin B (Sigma-Aldrich) were used as cell apoptosis (positive) controls. Results were expressed as the medial cytotoxic concentration ($CC_{50}$) using Prism GraphPad software for nonlinear regression analysis (https://www.graphpad.com/scientific-software/prism/).

**Antileishmanial activity in *L. (V.) panamensis* promastigotes.** We maintained exponential stage *L. (V.) panamensis* promastigotes (day 3 of culture) in RPMI 1640 medium supplemented with 10% FBS and cultured them in 96-well plates at a density of $5 \times 10^6$ parasites/mL. They were treated with quinoline alkaloid–like compounds at a concentration of 100, 10, 1, 0.1, and 0.01 μg/mL for 72 h at 26˚C. Next, we determined antileishmanial activity using the resazurin method (Sigma-Aldrich). Promastigotes treated with amphotericin B (Sigma-Aldrich) and pentamidine (Sanofi-Aventis, Gentilly, France) were used as parasite death (positive) controls. Results were expressed as the medial effective concentration ($EC_{50}$) using Prism

GraphPad software for nonlinear regression analysis (https://www.graphpad.com/scientific-software/prism/).

**Antileishmanial activity in *L. (V.) panamensis* intracellular amastigotes.** We cultured hMDMs at a density of $1 \times 10^6$ cells/mL in RPMI 1640 medium supplemented with 10% FBS. Next, 100 μL of the suspension was cultured in 96-well plates and infected with *L. (V.) panamensis* promastigotes (HMOM/COL/87/UA-140) at a 10:1 parasite:cell ratio. We incubated the plates in a 5% $CO_2$ atmosphere for 4 h at 35°C. After incubation, we removed noninternalized parasites by washing the cells with 0.9% NaCl saline solution (Baxter International, Deerfield, IL, USA). The cells were incubated again for 24 h and then treated with each quinoline alkaloid–like compound at a concentration of 100, 10, 1, 0.1, and 0.01 μg/mL in a 5% $CO_2$ atmosphere for 72 h at 36°C. Next, we quantified the percentage of infected cells using fluorescence microscopy, as previously described by Pérez-Cordero in 2011 [23], and quantified the parasite load (amount of intracellular promastigotes and amastigotes per cell) using Giemsa staining and flow cytometry (Cytomics FC 500 MPL; Beckman Coulter, Brea, CA, USA) [24]. Infected and incubated cells in RPMI 1640 medium were supplemented with 10% FBS and used as infection controls, while cells treated with different concentrations of amphotericin B (10, 1, 0.1, and 0.01 μg/mL) were used as treatment (positive) controls. Results were expressed as $EC_{50}$ using Prism GraphPad software's probit nonlinear regression analysis (https://www.graphpad.com/scientific-software/prism/).

## *In vivo* antileishmanial activity

**Evaluating the therapeutic *in vivo* response in hamsters.** We used a golden hamster (*Mesocrisetus auratus*) model to evaluate the therapeutic response of synthetic alkaloid compounds **2** and **8.** Male and female 6-week-old hamsters with a mean live weight (LW) of 120 g were infected by a dorsal intradermic injection of *L. (V.) panamensis* (MHOM/COL/87/UA140-EGFP) promastigotes. After the infected hamsters developed ulcers, they were randomly divided into three groups ($n = 6$ each) and treated topically with 1% ointment formulated from compounds **2** and **8**.

The hamsters were monitored every 4 weeks to assess their appearance and weight and the lesion size. We categorized each compound's effectiveness pre- and posttreatment as follows: (i) cure (complete disappearance of lesions), (ii) improvement (>10% reduced lesion area), and (iii) therapeutic failure or nonresponse (increased lesion size). The health status was monitored weekly for 3 months (12 weeks) posttreatment. After 3 months, the hamsters were euthanized in a $CO_2$ chamber. We performed histopathological analysis after necropsy of the affected tissues and organs. In addition, we evaluated treatment cytotoxicity using LW monitoring and biochemical parameters (alanine aminotransferase [ALT], creatinine, and blood urea nitrogen [BUN]) pretreatment and 45 days after ointment administration.

**Inducing apoptosis in hMDMs.** We cultured *L. (V.) panamensis*-infected and *L. (V.) panamensis*–noninfected hMDMs in 24-well plates at a density of $2 \times 10^6$ cells/mL in RPMI 1640 medium and treated them with quinoline alkaloid–like compounds in a 5% $CO_2$ atmosphere for 72 h at 35°C. Next, the cells were fixed with 4% paraformaldehyde for evaluating dead cells using a live/dead cell viability kit (Beckton, Dickinson and Company, Franklin Lakes, NJ, USA). hMDMs cultured without any treatment were used as viability controls, while those treated with 0.01% $H_2O_2$, 10 μg/mL of lipopolysaccharide (LPS; Sigma-Aldrich), and 1% dimethyl sulfoxide (DMSO) were used as cell mortality (positive) controls. Apoptotic and necrotic cells were quantified using flow cytometry (FACS Canto II; Becton, Dickinson and Company).

**Reactive oxygen species (ROS) and nitric oxide (NO) production in hMDMs.** We cultured hMDMs in 48-well plates in RPMI 1640 medium enriched with 10% FBS and 100 U/

100 μg/mL of 1% penicillin-streptomycin in a 5% $CO_2$ atmosphere for 72 h at 37°C. Next, the cells were infected with *L. (V.) panamensis* promastigotes at a 10:1 parasite:cell ratio for 24 h and then infected with quinoline alkaloid–like compounds at $EC_{50}$ with TI >3 (compounds **1**, **2**, and **8**). Uninfected cells were also incubated for 72 h in a 5% $CO_2$ atmosphere at 36°C and then treated with equivalent $EC_{50}$ concentrations of compounds **1**, **2**, and **8**. Next, we detached the cells with ethylenediaminetetraacetic acid (EDTA)/trypsin (Lonza, MD, USA) and 2',7'-dichlorodihydro-fluorescein diacetate ($H_2$DCFDA) tagged on a 5 μM fluorescent tube (Sigma-Aldrich) in order to identify intracellular ROS and/or 4-amino-5-methylamino-2′,7′-difluorofluorescein (DAF-FM; Sigma-Aldrich) for NO production in a 5% $CO_2$ atmosphere for 45 min at 37°C. ROS and NO were quantified using flow cytometry (FACS Canto II) at a wavelength of 480 nm. Untreated cells were used as negative controls, while cells treated with 0.0001% $H_2O_2$ and 1 μM phorbol myristate acetate were used as ROS production (positive) controls. LPS (10 ng/mL) [25] and phytohemag-glutinin (PHA-M) (10 μL/mL) (Invitrogen) were used as positive controls for NO production.

**Transmission electron microscopy of *L. (V) panamensis*-infected hMDMs.** We cultured hMDMs at a density of $1 \times 10^6$ cells/mL in RPMI 1640 medium supplemented with 10% FBS in a 5% $CO_2$ atmosphere for 72 h at 37°C and then infected them with *L. (V.) panamensis* promasti-gotes in the stationary stage at a 10:1 parasite:cell ratio in a 5% $CO_2$ atmosphere for 4 h at 36°C. Next, we removed non-internalized parasites, incubated the infected cells again in $CO_2$ for 24 h at 36°C, exposed the cells at $EC_{50}$ for compounds **1**, **2**, and **8** in a 5% $CO_2$ atmosphere for 72 h at 36°C, fixed the cells with 2.5% glutaraldehyde, and visualized them using transmission electron microscopy (TEM). This procedure involved including fresh resin and 100–200-nm-wide ultra-fine cuts observed by TEM (100X–10.000X) (HITACHI-HU-12; Hitachi, Tokyo, Japan).

## Statistical analysis

All assays required at least three independently duplicated experiments. We determined cyto-toxicity against hMDMs and effectivity in *L. (V) panamensis* promastigotes according to each treatment's mortality (compound and concentration) using Eq 1, where the control mean fluo-rescence intensity (MFI) was 100% viability:

$$\text{Validity inhibition } (\%) = 100 - \left[ \frac{\text{MIF exposed cells}}{\text{MIF unexposed cells}} \times 100 \right] \tag{1}$$

We used GraphPad Prism nonlinear regression analysis to estimate the percentage mortal-ity with regard to $CC_{50}$ and $EC_{50}$. Cytotoxicity was classified as high if $CC_{50} < 100$ μg/mL, moderate if $CC_{50}$ from 100 to <200 μg/mL, or low if $CC_{50} > = 200$ μg/mL, depending on each compound's induced effect. The antileishmanial activity with regard to promastigotes was clas-sified as active according to Eq 2:

$$\text{Active} = EC50 < \left[ \frac{LC50}{2} \right], \tag{2}$$

where $LC_{50}$ is the median lethal concentration. We also used GraphPad Prism nonlinear regression analysis to determine the intracellular antileishmanial activity (concentration vs. response), depending on the infection (number of infected cells) and parasite burden (number of parasites/cell) obtained in experimental conditions. Parasite inhibition was estimated using Eq 3, where the plate MFI is 100% parasites:

$$\text{Infection inhibition } (\%) = 100 - \left[ \left( \frac{\text{MIF exposed parasites}}{\text{MIF unexposed parasites}} \times 100 \right) \right] \tag{3}$$

The percentage inhibition was used to calculate $EC_{50}$ by nonlinear regression analysis (fluorescence microscopy), while the parasite burden was evaluated as the percentage of intracellular promastigotes and amastigotes in 100 infected cells (Giemsa staining). The antileishmanial activity was classified as high if $EC_{50} < 25$ μg/mL, moderate if $EC_{50}$ from 25 to <50 μg/mL, and low if $EC_{50} >= 50$ μg/mL. The selectivity index (SI) was estimated using the following formula:

$$SI = \frac{CC_{50}}{EC_{50}}$$

The *in vivo* efficacy of compounds **1**, **2**, and **8** was expressed as a percentage cure, improvement, or nonresponse compared to the beginning of treatment. Efficacy was evaluated *in vivo* to compare percentages, the frequency of change, and treatment cytotoxicity in the hamster model. We used GraphPad Prism to determine ROS and NO production and the apoptotic cell percentage. One- or two-way analysis of variance (ANOVA; response–time kinetics) was used to compare treatments after confirming normal data distribution. $P < 0.05$ was considered statistically significant by comparing untreated cells (experiments on untreated hMDMs) with infected cells (experiments on untreated and treated hMDMs).

## Ethics statement

This study was conducted in strict accordance with the recommendations by the *Universidad de Antioquia's Guide for the Care and Use of Laboratory Animals* and the Universidad Nacional de Colombia's Ethics Committee with regard to the use of human cells (document 04–2017). The protocol was approved by the Universidad de Antioquia's Animal Experiments Ethics Committee (protocol name "Comprehensive Leishmaniasis Control Strategy in Colombia").

## Results

### Synthetic analogs of compound 1

We characterized seven quinoline alkaloid–type compounds similar to compound **1** (template) for having 1-azanaphthalene in their central skeleton and substituents in their carbon structure.

Compounds **1** and **8** had similar characteristics to the quinoline heterocyclic structure in C-5. However, there were structural differences between compound **1** and the other quinoline alkaloid–type compounds in the oxygenated substituents (OH) in the reference structure compared to compound **2** (Table 1). Compounds **3**, **5**, **6**, and **8** lacked one heterocycle compared to compound **1**. The imidazole and amino nitrogen groups of compounds **7** and **8** were the least similar to the comparison pattern, indicating heterogeneity.

The physicochemical properties of these compounds gave permeation coefficients log$P$ >1 (neutral state), with weakly similar compounds having the lowest log$P$ compared to compounds like compound **1**. In addition, no significant changes were identified regarding previously estimated log$P$ values when predicting the permeability coefficients of ionized subspecies (log$D$, pH 7.2). PSA analysis, related to a compound's capability of interacting in a polar environment, showed that the compounds had a PSA of <60 [26].

The compounds were predicted to be soluble or moderately soluble regardless of their permeation capacity (log$P$ and log$D$).

### *In vitro* cytotoxicity and antileishmanial activity

Most of the tested compounds (75%) had high cytotoxicity toward hMDMs, with $CC_{50} <$ 100 μg/mL. Only compounds **3** and **5** had moderate cytotoxicity ($CC_{50} > 100$ μg/mL).

**Table 1. Synthetic and natural quinoline alkaloid compounds' structural and physicochemical properties.**

| Name | Compound | TI | Molecular weight (g/mol) | LogP | LogD pH 7.4 | PSA | Solubility |
|------|----------|-----|--------------------------|------|-------------|-----|------------|
| 1 | N-methyl-8-methoxyflindersin | N.A | 271,32 | 2,65 | 2,73 | 31,3 | Moderate |
| 2 | 1,2,3,4-tetrahydro(Benzo)-3-quinolin-ol | 0.731 | 199,24 | 1,97 | 2,63 | 32,26 | Moderate |
| 3 | Carboxylic acid 2-ethyl-3-propyl-4-quinoline | 0.724 | 243,3 | 2,86 | 2,04 | 32,98 | Soluble |
| 4 | 4-methyl-2-(1H)-quinoline | 0.791 | 159,1 | 2,41 | 2,33 | 32,86 | Moderate |
| 5 | 4,7,8-trimetoxifure [2,3-b]-quinolin | 0.821 | 243,3 | 2,58 | 2,33 | 53,7 | Moderate |
| 6 | Carboxylic acid 7,8-tetrahydro-quinoline | 0.8 | 307,3 | 3,4 | 1,65 | 59,4 | Moderate |
| 7 | 3,4-dimethyl-3H-imidazol [4.5f]- quinolin-2-amine | 0.67 | 198,2 | 1,41 | 1,73 | 56,7 | Soluble |
| 8 | 2-amino-8-hidroxiquinoline | 0.704 | 160,7 | 1,32 | 1,09 | 59,14 | Soluble |

Physicochemical properties (lipophilicity, flexibility, insaturation, solubility, polarity, size, and molecular weight) of quinoline alkaloid compounds: 2D chemical structure of evaluated compounds.

LogP/LogD (lipophilicity).

TI, Tanimoto index; PSA, polar surface area.

Between-group comparison showed no correlation with regard to the compounds' toxic concentrations for hMDMs (Table 2).

The antileishmanial activity was determined for intracellular amastigotes. Compounds **1**, **2**, and **8** had high antileishmanial activity, with $EC_{50} < 10$ μg/mL (50 μM). The SI calculated from cytotoxicity correlation and antileishmanial activity was high for compounds **1** (5.87 SI), **2** (3.64), and **8** (5.44) (Table 2). We selected these three compounds for evaluating antileishmanial activity using infected hMDMs, in addition to ultrastructural alterations and therapeutic effects *in vivo*.

## Evaluating antileishmanial activity *in vivo*

We evaluated clinical parameters and treatment cytotoxicity in hamsters experimentally infected by *L. (V.) panamensis* in order to determine the selected compounds' therapeutic

**Table 2. Synthetic and natural quinoline alkaloid–like compounds' biological activity.**

| Compounds | TI[a] | CC50 μg/mL (μM) | EC50 μg/mL (μM) | | SI[b] |
|-----------|-------|-----------------|-----------------|-----|-------|
| | | | Intracellular promastigotes | Intracellular amastigotes | |
| 1 | N.A | 43.6 ± 3.2 (161 ± 11.8) | 9.9 ± 3 (36.5 ± 1) | 7.4 ± 1.95 (29.2 ± 5.3) | 6.25 |
| 2 | 0.73 | 18.9 ± 0.9 (95 ± 4.5) | 3.4 ± 1.1 (17 ± 5.3) | 5.2 ± 0.9 (26 ± 4.8) | **3.63** |
| 3 | 0.72 | 155 ± 6.1 (817 ± 32.3) | 93.8 ± 6.3 (495.6 ± 33.3) | 121.9 ± 6.2 (495.6 ± 32.6) | 1.27 |
| 4 | 0.79 | 95.2 ± 3.3 (391 ± 13.4) | 62.3 ± 5.3 (256.1 ± 21.9) | 57.1 ± 5.0 (256.1 ± 20.7) | 1.66 |
| 5 | 0.82 | 107 ± 7 (411 ± 27) | 66.4 ± 13 (256.2 ± 31.5) | 61.7 ± 1.1 (256.2 ± 4.3) | 1.73 |
| 6 | 0.81 | 88 ± 13 (344 ± 52.2) | 61.3 ± 29.6 (239.7 ± 105) | 46.5 ± 9.4 (239.7 ± 36.6) | 1.89 |
| 7 | 0.67 | 95.2 ± 4.3 (480 ± 16.4) | 26.5 ± 5.8 (133.7 ± 29.2) | 55.8 ± 8.3 (133.7 ± 42.1) | 1.70 |
| 8 | 0.7 | 5.5 ± 1.1 (34 ± 6.9) | 1.6 ± 0.8 (10 ± 4.7) | 1.0 ± 0.4 (6.6 ± 2.6) | **5.5** |
| AMB[c] | - | 4.5 ± 0.6 (4.9 ± 0.7) | 0.3 ± 0.2 (0.3 ± 0.2) | 0.4 ± 0.1 (0.4 ± 0.1) | 11.25 |

The data represent the $CC_{50}$ for each compound ± SD evaluated in hMDMs, $EC_{50}$ (μM) for each compound ± SD in *Leishmania (Viannia) panamensis* (promastigotes) and $EC_{50}$ for each compound ± SD in *L. (V.) panamensis* intracellular amastigotes.

[a]TI, Tanimoto index.

[b]SI, selectivity index = $CC_{50}/EC_{50}$.

AMB, amphotericin B; $CC_{50}$, medial cytotoxic concentration; $EC_{50}$, medial effective concentration; SD, standard deviation.

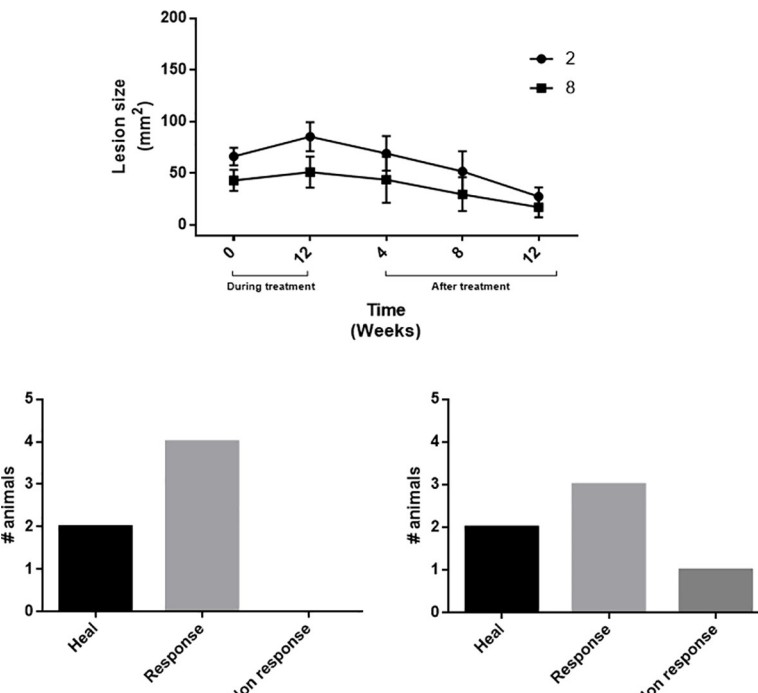

**Fig 1. Effect of treatment with compounds 2 and 8 on hamsters suffering from CL.** (a) Lesion size measurement (mm$^2$) during treatment (pre- and posttreatment: weeks 4–12). Number of recovered hamsters showing improvement or lacking response to treatment with (b) compound **2** and (c) compound **8** (*n* = 6 for each treatment). CL, cutaneous leishmaniasis.

effects. Fig 1 shows the evolution of lesions. Treatment with compounds **2** and **8** significantly reduced lesions during treatment (Fig 1A). Compound **2** induced total cure (healing) with regard to lesion size 3 months posttreatment in 33.3% of the experimental group, and the remaining 66.6% improved satisfactorily, showing a reduced lesion size. Compound **8** induced total reduction of lesions in 33.3% of the experimental group, clinical improvement in 50%, and no effect (poor response) in 16.6% (Fig 1B and 1C)

Histopathologic analysis of the skin of hamsters that were cured by treatment with compound **2** showed moderate orthokeratotic hyperkeratosis and low mononuclear leukocyte infiltration (Fig 2A), while hamsters showing improvement had severe leukocyte infiltratation and abundant *L. (V.) panamensis* being phagocytosed by hMDMs (Fig 2B). We observed mixed leucocyte infiltratation in hamsters cured by treatment with compound **8**, showing a predominance of neutrophils and hMDMs phagocytosing few *L. (V.) panamensis* (Fig 2D).

No cytotoxicity-related clinical manifestations were observed with regard to treatment with compound **2** or **8**. Fig 3A shows the stability of the experimental group's weight posttreatment with compounds **2** and **8**, that is, there were no evident changes during treatment.

Blood chemistry analysis for each hamster from the experimental group showed normal creatinine and ALT levels (Fig 3B–3D), while BUN levels in hamsters treated with compound **8** significantly decreased (*P* < 0.005) compared to before treatment (Fig 3C).

## Biochemical (NO and ROS production) and structural changes in infected hMDMs

### Cell apoptosis in *L. (V.) panamensis*-infected hMDMs.
We performed flow cytometry to evaluate cell apoptosis induction in infected hMDMs and those hMDMs treated with

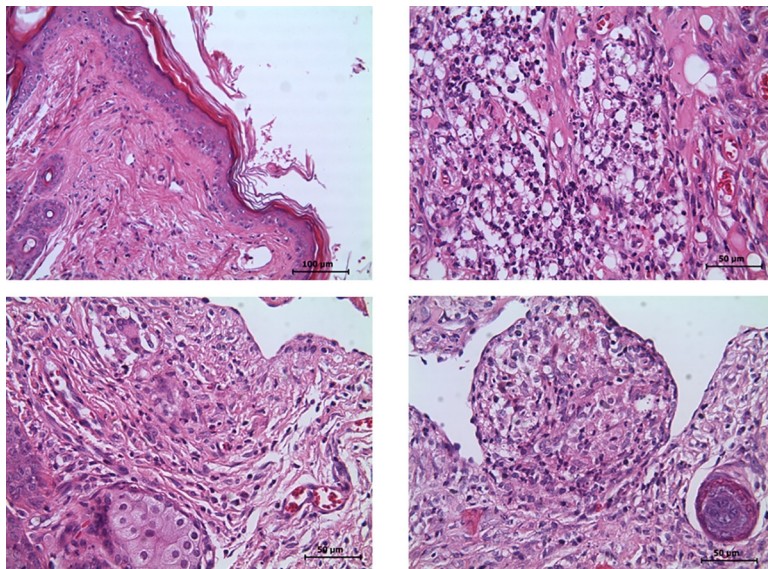

**Fig 2. Histopathology of the skin of infected hamsters treated with compounds 2 and 8.** (a) Skin of cured hamsters treated with compound **2**, (b) skin showing improvement, and skin of hamsters treated with compound **8**, showing (c) cure or (d) improvement: hematoxylin/eosin method; 50–100 μm magnification.

compounds **1**, **2**, and **8**, considering that inducing cell apoptosis in host cells for parasites can trigger intracellular parasite elimination or multiplication [27, 28]. *L. (V.) panamensis* infection significantly increased the number of apoptotic cells compared to noninfected hMDMs. Posttreatment (Fig 4), compound **1** significantly increased the percentage of apoptotic infected cells (41.3%) compared to basal cells with the same treatment (20.2%) ($p < 0.005$), while compound **2** did not evidently induce inducing apoptosis of infected and noninfected hMDMs ($p < 0.005$), and compound **8** significantly increased the percentage of $V^+$ $7AAD^-$annexin cells (late apoptosis) (58.03%) compared to infected hMDMs (45.25%) ($p < 0.005$), having a direct effect on host cells.

**ROS and NO expression in *L. (V) panamensis-infected* hMDMs.** ROS production in hMDMs treated with compounds **1**, **2**, and **8** after 4, 24, 48, and 72 h of treatment was evaluated because modulating the innate immune response through parasiticidal molecules plays an important role in controlling infection progression (Fig 2). *L. (V.) panamensis* did not induce an increase in ROS generation during the first 4 h postinfection; however, infected hMDMs treated with compound **2** showed an increase in ROS generation. In addition, no significant changes in oxidative breakdown in infected and treated cells were observed 24 h posttreatment. However, *L. (V.) panamensis* infection induced ROS generation 48 h posttreatment, and infected hMDMs treated with compound **2** showed a decrease in ROS generation (Fig 5). There were no significant variations with regard to infected hMDMs treated with compounds **1** and **8** compared to infected control cells.

NO production in infected hMDMs treated with compounds **1**, **2**, and **8** was evaluated because its production in hMDMs plays an important role in CL control. *L. (V.) panamensis* infection did not significantly increase intracellular NO levels compared to uninfected hMDMs (Fig 6). However, hMDMs treated with compound **1** showed an increase in NO production compared to untreated infected hMDMs (Fig 6). Synthetic compounds did not induce this pattern.

NO production kinetics in infected hMDMs 24, 48, and 72 h posttreatment were evaluated because NO production in hMDMs increased after phagocytosis and oxidative breakdown

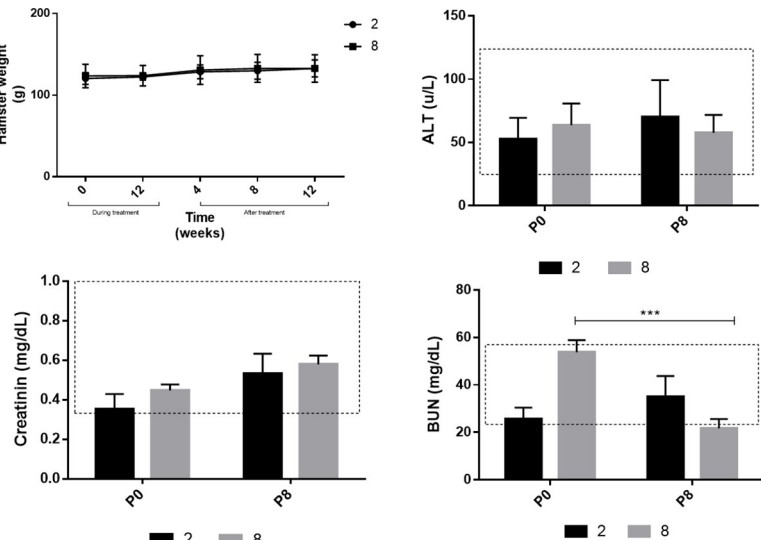

**Fig 3. LW (g) and serum ALT, creatinine, and BUN levels pre and posttreatment with compounds 2 and 8.** Data
are presented as mean ± SD of (a) LW, (b) ALT, (c) creatinine, and (d) BUN in the serum of hamsters suffering from
CL before (P0) and after (P8) treatment. Significant between-group differences ($p < 0.05$). The dotted area represents
the reference values for each parameter. LW, live weight; ALT, alanine aminotransferase; BUN, blood urea nitrogen;
SD, standard deviation; CL, cutaneous leishmaniasis.

(Fig 6). No significant changes in NO production in *L. (V) panamensis*-infected hMDMs were
observed 48 h posttreatment; however, we found an increase in NO production in infected
hMDMs 72 h posttreatment with compound **1** (Fig 6).

## *L. (V.) panamensis*-infected hMDM ultrastructural alterations

We evaluated ultrastructural alterations in infected hMDMs treated with compounds **1**, **2**, and **8**
after 72 h of treatment by TEM to ascertain the effects of compounds **1**, **2**, and **8** on host cell
internal structures. We compiled evident changes according to morphological characteristics
(Table 3). Infected hMDMs had intact nuclei with a normal chromatin structure, intact cell mem-
branes, autophagic vacuoles (vacuoles having cytoplasmatic content and a double membrane),
and several pseudopods and mitochondria but no lipid bodies (LB; electrodense vacuoles). We
also observed several parasites inside the parasitophorous vacuoles (PV) (Fig 7A and 7B).

Treatment with compound **1** resulted in an intact nucleus, several autophagic vacuoles, and
limited intracellular promastigotes and amastigotes surrounded by individual parasitic vacu-
oles and double-membrane vacuoles (autophagosomes) (Fig 7C and 7D).

Infected hMDMs treated with compound **2** had nuclei with normal chromatin distribution,
no double-membrane vacuoles, and several LB in the cytoplasm. PV had parasites inside them
(Fig 7E and 7F) in contrast to infected and untreated hMDMs (Fig 7A and 7B). Infected
hMDMs treated with alkaloid **8** had altered nucleus integrity (nuclear membrane), a lack of
LB, several double-membrane vacuoles, and a few intracellular promastigotes and amastigotes
in individual PV surrounding intracellular amastigotes (Fig 7G and 7H).

## Ultrastructural modifications of quinoline alkaloid–like compounds in *L. (V) panamensis* intracellular amastigotes

We determined the effects of quinoline alkaloid–like compounds on *L. (V.) panamensis* intra-
cellular amastigotes, as previously described (Fig 8 and Table 4). Intracellular amastigotes in

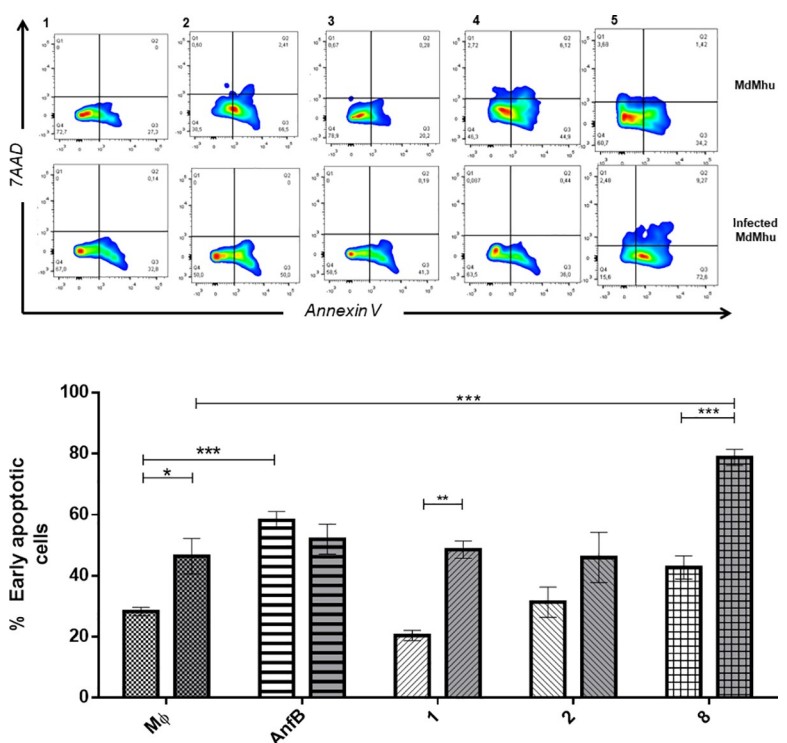

**Fig 4. Percentage hMDMs during early apoptosis (annexin V⁺ 7AAD⁻). hMDMs treated with quinoline alkaloid–like compounds at EC₅₀, evaluated by flow cytometry.** (A) Dot plot of noninfected hMDMs: *x* axis, annexin V-PE; *y* axis, 7AAD. (1) M (uninfected or untreated hMDMs); (2) AmpB; (3) compound **1**; (4) compound **2**; (5) compound **8**. (B) Bar plot of the percentage of events (cells) for (a) noninfected hMDMs and (b) *Leishmania (Viannia) panamensis*-infected hMDMs (IMs) treated with compounds **1**, **2**, and **8**. *$p < 0.05$; **$p < 0.01$; ***$p < 0.005$. EC₅₀, medial effective concentration; hMDMs, human monocyte-derived macrophages.

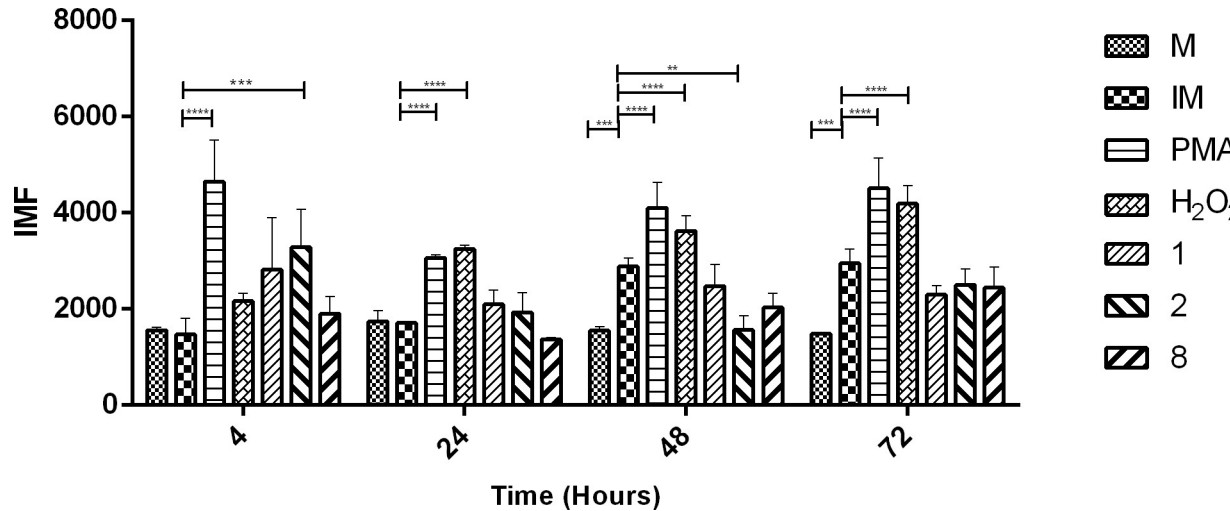

**Fig 5. ROS generation in *Leishmania (Viannia) panamensis-infected* hMDMs treated with quinoline alkaloid–like compounds at different times (h).** MFI. Comparing ROS induction by compounds in hMDMs. *$p < 0.05$; **$p < 0.01$; ***$p < 0.005$. ROS, reactive oxygen species; hMDMs, human monocyte-derived macrophages; MFI, medium fluorescent intensity; M, hMDMs; IM, infected hMDMs.

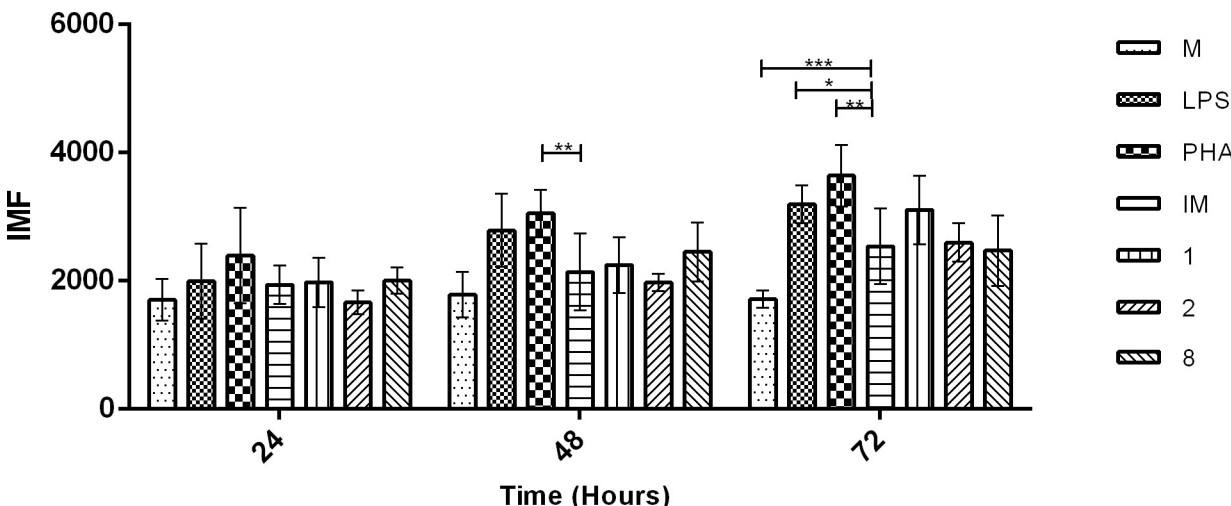

**Fig 6. Inducing NO production in *L. (V) panamensis*-infected hMDM.** NO production kinetics in hMDMs infected with *L. (V.) panamensis* and treated with quinoline alkaloid–like compounds **1**, **2**, and **8** after 24, 48, and 72 h of treatment. $^*p < 0.05$; $^{**}p < 0.01$; $^{***}p < 0.005$. NO, nitric oxide; hMDMs, human monocyte-derived macrophages; M, uninfected or untreated hMDMs; LPS, lipopolysaccharide; IM, infected hMDMs.

untreated hMDMs had nuclei with normal chromatin distribution, entire cell membranes (Fig 8A and 8B), kinetoplasts and mitochondrial membranes without evident alterations (Fig 8A), and individual PV and no evidence of apparent cytoplasmatic damage (Fig 8A and 8B).

Intracellular amastigotes in hMDMs treated with compound **1** lost their cell membrane integrity (Fig 8C and 8D), together with acidocalcisomes (electrodense structures) having electron-dense structures (Fig 8D). We also observed a loss of intracellular amastigotes in the cell membrane (Fig 8D). Intracellular promastigotes and amastigotes in hMDMs treated with compound **2** had nuclei with abnormal chromatin distribution, cytoplasm vacuolization, multiorganelles (two kinetoplasts or two axonemes), and LB in the cytoplasm (Fig 8E and 8F). Intracellular amastigotes in hMDMs treated with compound **8** had a vacuolated cytoplasm,

**Table 3. Ultrastructural modifications of *Leishmania (Viannia) panamensis*-infected hMDMs treated with quinoline alkaloid–like compounds.**

| Treatment | Nucleus | Cell membrane | Vacuoles | Mitochondria | LB | PV | Parasites |
|---|---|---|---|---|---|---|---|
| **nTIC** | - | - | ++ | +++ | - | +++ | +++ |
| | | Presence of pseudopods | Double membrane (autophagic-digestives) | | | Individuals, attached to the parasite | |
| **1** | - | - | ++ | + | - | + | + |
| | | | Double membrane (autophagic-digestives) | | | Individuals, attached to the parasite | |
| **2** | - | - | - | + | +++ | + | ++ |
| | | | | | | Multiple parasites per vacuole | |
| **8** | +++ | - | +++ | + | - | + | + |
| | Abnormal distribution of chromatin | | Double membrane (autophagic-digestives) | | | Individuals, attached to the parasite | |
| | Nuclear membrane alteration | | | | | | |

(-) no alteration/absence; (+) low alteration/low presence; (++) alteration/moderate presence; (+++) serious disturbance/abundant presence.

hMDMs, human monocyte-derived macrophages; nTP, untreated parasites; LB, lipid bodies; PV, parasitophorous vacuoles.

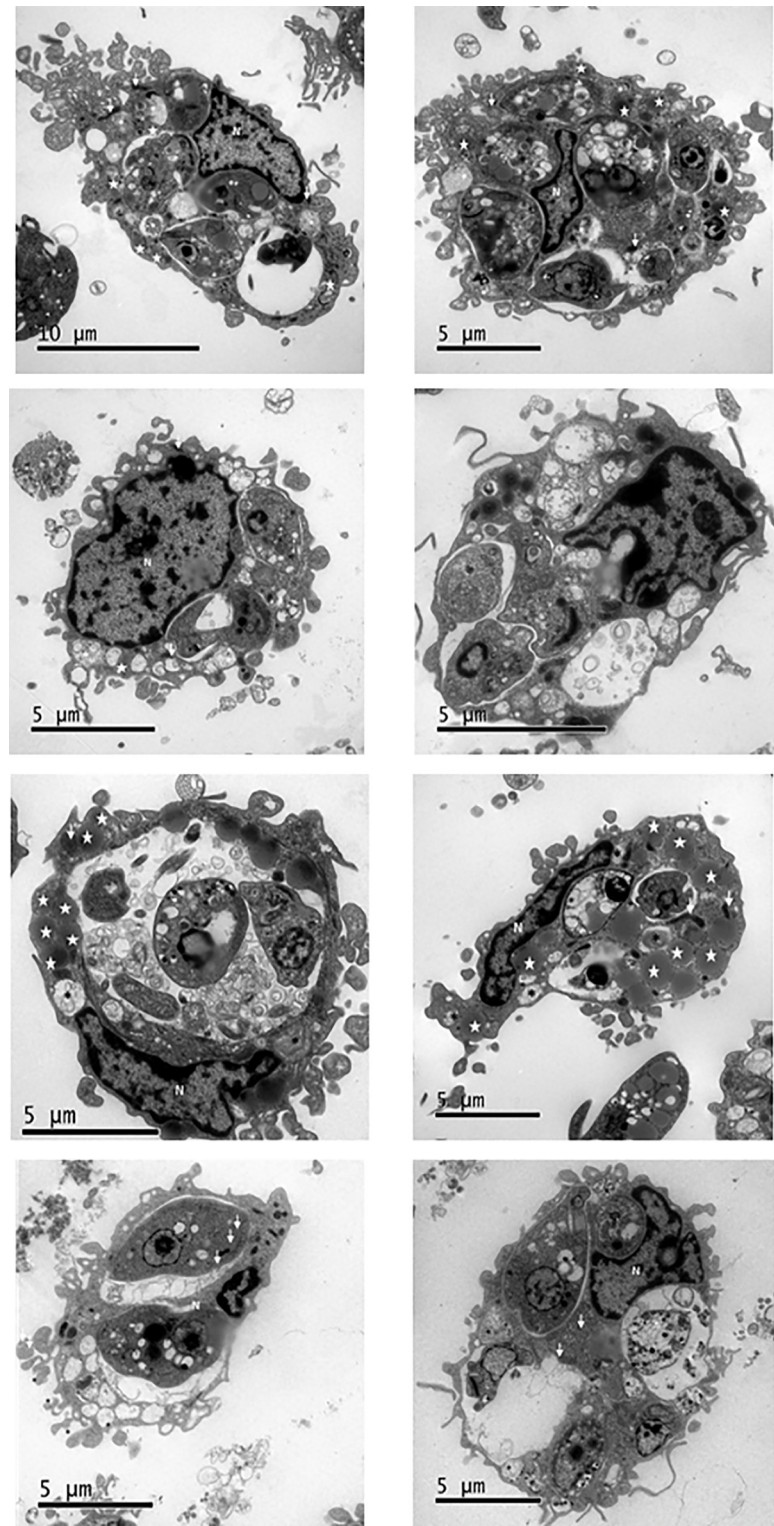

**Fig 7. Alterations in *Leishmania (Viannia) panamensis*-infected hMDMs' ultrastructures.** Untreated infected hMDMs (a and b). Infected hMDMs treated with compounds (c and d) **1**, (e and f) **2**, and (g and h) **8**. White arrows, mitochondria; white asterisks, double-membrane vacuoles (autophagosomes); white stars, electrodense bodies. Visualization scale = 5–10 μm; 10,000x magnification. hMDMs, human monocyte-derived macrophages; N, nucleus; PV, parasitophorous vacuole; TEM, transmission electron microscopy.

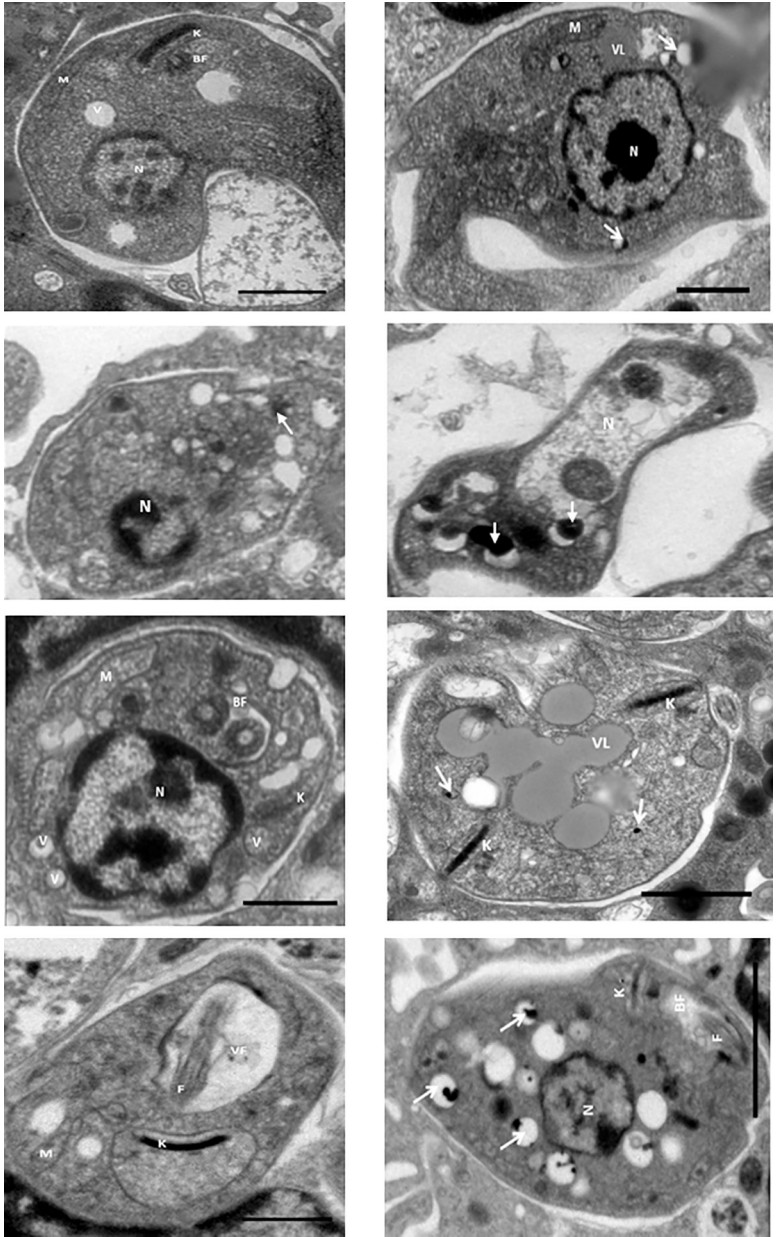

**Fig 8. Ultrastructural alterations of *Leishmania (Viannia) panamensis* intracellular amastigotes.** Untreated hMDMs (a and b). hMDMs treated with compounds (c and d) **1**, (e and f) **2**, and (g and h) **8**. White arrows, acidocalcisomes. Visualization scale = 2 μm; 10,000X magnification. hMDMs, human monocyte-derived macrophages; N, nucleus; F, flagellae; K, kinetoplast; V, vacuole; LV, lipid vacuole (electrodense vacuoles); M, mitochondria; TEM, transmission electron microscopy.

acidocalcisomes, and widening of the kinetoplast membrane, mitochondria, and flagellar pockets (Fig 8G and 8H).

## Discussion

Leishmaniasis is a parasitic disease affecting more than 98 countries. Most CL cases are found in Afghanistan, Algeria, Brazil, the Islamic Republic of Iran, Pakistan, Peru, Saudi Arabia, the

**Table 4. Ultrastructural alterations of *Leishmania (Viannia) panamensis* intracellular amastigotes in hMDMs treated with quinoline alkaloid–like compounds.**

| Treatment | Nucleus | Cell membrane | Vacuoles | Mitochondria/ Kinetoplasts | LB | Flagellum (flagellar/ axonema pocket) | PV |
|---|---|---|---|---|---|---|---|
| nTP | - | - | - | - | - | - | ++ |
| | | | | | | | Individuals attached to the parasite |
| 1 | +++ | +++ | ++ | - | - | - | + |
| | Nuclear membrane damage | Integrity damage in membrane | Electrodense acidocalcisome type | | | | Individuals attached to the parasite |
| 2 | ++ | - | +++ | ++ | ++ + | ++ | + |
| | Abnormal distribution of chromatine | | Double membrane (autophagic-digestives) | Multiple organelles | | Multiple organelles | Multiple |
| 8 | ++ | - | +++ | +++ | - | +++ | + |
| | Abnormal distribution of chromatine | | Double membrane (autophagic-digestives) | Membrane swelling | | Membrane swelling | Individuals attached to the parasite |

(-) no alteration/absence; (+) low alteration/low presence; (++) alteration/moderate presence; (+++) serious disturbance/abundant presence.

hMDMs, human monocyte-derived macrophages; nTP, nontreated parasites; LB, lipid bodies; PV, parasitophorous vacuoles.

Syrian Arab Republic, and Colombia [1, 29–31]. The current treatment has adverse hepatic, cardiac, and teratogenic effects [6, 32], which, together with patients abandoning treatment, contribute to the emergence of drug-resistant parasites [33].

Many quinoline alkaloids, including *N*-methyl-8-methoxyflindersine (**1**), show antileishmanial activity *in vitro* [15–18, 34], and we tested synthetic analogs of compound **1** to identify antileishmanial compounds. Seven compounds share a quinoline structure with compound **1**, although there are differences in oxygen and nitrogen substituents. Like compound **1**, compounds **2** and **8** show especially high cytotoxicity toward human cells. However, their biological activity against *L. (V.) panamensis* intracellular amastigotes has SI >3.

*In vivo* therapeutic validation in hamsters shows that compound **2** has a 40% healing rate during re-epithelialization, while clinical improvement in hamsters (60%) involves typical chronic inflammation and parasitic presence. Compound **8** has a 50% cure rate, accompanied by neutrophil and macrophage migration and a few intracellular amastigotes. Hamsters that lack a response to treatment can partially control the lesion size, indicating that the compounds can control tissue damage caused by the parasite. Compounds **2** and **8** do not induce obvious changes with regard to cytotoxicity parameters, indicating that the compounds may be safe therapeutic candidates for treating CL.

Compound cytotoxicity can trigger cell apoptosis in host hMDMs. Although compound **8** treatment tends to increase apoptotic cells compared to infected, untreated, and treated hMDMs, none of the evaluated compounds significantly induces an increase in apoptotic or necrotic cells.

Host cell apoptosis modulation is a strategy used against *Leishmania* inside a host [35–37]. *L. (V.). panamensis*-infected hMDMs show a significant increase in the number of apoptotic cells 72 h posttreatment, related to double-membrane vacuoles in the cytoplasm, blisters on the cell membrane, and preserved nuclei in the hMDMs. Apoptosis induced in *L. (V). panamensis*-infected hMDMs has a differential behavior with regard to other parasites from the *Viannia* genus, increasing necrosis in C57BL/6 and BALB/c mouse hMDMs [38–40]. Infected hMDMs treated with compound **8** show a significant increase in apoptotic cells 72 h posttreatment, together with numerous autophagosomes and the preservation of cell and nuclear membrane integrity in the hMDMs [41–43].

The immune response is an essential component in CL control, and treatment modulating this response could contribute toward resolving CL [44, 45]. Compound **2** induces significant ROS generation during the first hours where phagocytosis is involved, in addition to remodeling the hMDM cytoskeleton [38, 46]. However, 48 hours posttreatment, a reduction in oxidative burst can be seen, indicating a metabolic change in hMDMs, that is, resolution of the infection phase [42, 46]. Given that compound **8** does not have similar effects as compound **2**, the quinoline ring is not the only structure responsible for treatment effects.

Although oxidative response and ROS generation by hMDMs contribute to CL control [28, 36], NO production is most important in parasite elimination [47]. Compound **1** increases NO production 72 h posttreatment, which is consistent with studies in which compound **1** was unable to induce NO after 48 h in antigen-presenting cells [17]. In addition, *L. donovani*–infected murine macrophages in BALB/c mice treated with quaternary quinolines show a significant increase in NO production [43]. This effect confirms that some quinoline alkaloids, like compound **1**, can have a dual function (immunomodulating hMDMs and having a direct effect on parasites) and that such activity depends on the type of cells used for antiparasitic evaluation. Compounds **2** and **8** do not modulate the induction of NO production in hMDMs, indicating that compound **2** can only modulate intracellular pathways involved in oxidative burst within hMDMs and that compound **8** can exert direct effects on intracellular promastigotes and amastigotes [47–49].

The apparent mechanism of action (MoA) of compounds **1**, **2**, and **8** is related to the effect on intracellular promastigotes and amastigotes. Compound **1** causes damage to the parasitic cell membrane, indicating that compound **1**'s action can cause intracellular amastigote death by inducing a loss of cell membrane integrity due to nitrogenous species produced by hMDMs 72 h posttreatment [50]. Compound **2** induces apoptosis in intracellular amastigotes [51], indicating an MoA targeting apoptosis-related pathways [50, 52], possibly because of hMDMs' oxidative response during the first few hours of treatment. Compound **8** induces cell apoptosis in intracellular promastigotes and amastigotes [38, 51, 53], indicating that it may inhibit parasitic growth and development (evidenced by a decreased parasitic load) by inducing apoptosis in exclusively infected hMDMs.

Our results showed that synthetic compounds **2** and **8** show antileishmanial activity against both parasitic stages (promastigotes and amastigotes), and compound **2** induces oxidative burst in hMDMs, with a significant bearing on improving treatment for and curing CL. The search strategy for compounds showing antiparasitic activity and structural similarity identified two synthetic compounds with similar activity as that shown by a naturally occurring metabolite (used as a base compound), differing with regard to oxygen and nitrogen substitutes or free radicals in their structure [12, 13]. This suggests that an *in silico* strategy could help find new alternatives to overcome the limitations in developing new medicines from natural molecules and facilitate the continuation of drug development in preclinical or clinical studies.

## Acknowledgments

We thank the IDCBIS (District Institute of Science, Technology, and Innovation in Health) in Bogota, Colombia; for their kind donation of Buffy coat units to carry out this project.

## Author Contributions

**Conceptualization:** Elaine Torres Suarez, Gabriela Delgado.

**Formal analysis:** Elaine Torres Suarez, Gabriela Delgado.

**Funding acquisition:** Elaine Torres Suarez, Diana Susana Granados-Falla, Gabriela Delgado.

**Investigation:** Elaine Torres Suarez.

**Methodology:** Elaine Torres Suarez, Sara María Robledo, Javier Murillo, Yulieth Upegui.

**Project administration:** Elaine Torres Suarez, Gabriela Delgado.

**Supervision:** Diana Susana Granados-Falla, Sara María Robledo, Gabriela Delgado.

**Writing – original draft:** Elaine Torres Suarez.

**Writing – review & editing:** Elaine Torres Suarez, Diana Susana Granados-Falla, Sara María Robledo, Gabriela Delgado.

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
