## [Decision Letter · Decision Letter 0]

31 Jul 2020

PONE-D-20-17746

Antileishmanial activity of synthetic quinolinic alkaloids on the cutaneous experimental leishmaniasis

PLOS ONE

Dear Dr. Delgado,

Thank you for submitting your manuscript to PLOS ONE. After careful consideration, we feel that it has merit but does not fully meet PLOS ONE’s publication criteria as it currently stands. Therefore, we invite you to submit a revised version of the manuscript that addresses the points raised during the review process.

Next to the revisions required by the reviewer (which must all be addressed and a point to point rebutal on how they have been addressed must be provided), I recommend that the manuscript will be reviewed by a native English speaking reviewer before resubmitting the manuscript. Provide evidence to demostrate that thias was done..

In addiiton, follow the guidelines for prepating a manuscript and pay particular attention to the references.

We look forward to receiving your revised manuscript.

Kind regards,

Henk D. F. H. Schallig, Ph.D

Academic Editor

PLOS ONE

Journal Requirements:

Reviewers' comments:

Reviewer's Responses to Questions

**Comments to the Author**

1. Is the manuscript technically sound, and do the data support the conclusions?

Reviewer #1: Yes

2. Has the statistical analysis been performed appropriately and rigorously? 

Reviewer #1: Yes

3. Have the authors made all data underlying the findings in their manuscript fully available?

Reviewer #1: Yes

4. Is the manuscript presented in an intelligible fashion and written in standard English?

Reviewer #1: No

5. Review Comments to the Author

Reviewer #1: This manuscript describes the antileishmanial effects of synthetic analogues of quinoline alkaloids from members of the Plant family Rutaceae, in particular N-methyl-8-methoxyflindersine. This compound has previously been found to display important antileishmanial activity in preclinical models but can only be obtained in small amounts from the natural sources and is difficult to synthesize. Using an in silico approach, eight analogues have been identified which have been evaluated for their effects against cultured Leishmania (V). panamensis promastigotes and amastigotes in cultured human macrophages. Subsequently, the two best performing analogues (2 and 8) were tested in golden hamsters with experimental cutaneous leishmaniasis caused by L. (V). panamensis promastigotes. In addition, various studies on the potential mechanism of action of the test compounds have been carried out. Based on the results obtained, it was concluded that compounds 2 and 8 may be pursued as treatment options of CL.

General comments

Considering the limited clinical efficacy, notable toxicity, and relatively high costs of the currently available forms of treatment of leishmaniasis including CL, there is an urgent need of improved and affordable medications against this disease. For this reason, this manuscript has merit, indicating the potential usefulness of synthetic quinolinic alkaloids against CL. The experiments have been well selected and carried out, and the results are convincing and support the conclusion.

However, the authors must make a number of important corrections before the manuscript can be published.

1. First of all, the use of the English language is often faulty and spoils the readability of the manuscript. One of the many errors is the spelling of ‘N-metil-8-metoxiflindersin’ that should be ‘N-methyl-8-methoxyflindersin’. The authors are advised to have a native English speaker review the manuscript before resubmitting it.

2. Secondly, the authors should try to write up the Discussion more concisely; the lengthy wording distracts from the message they want to convey. One way to go ahead is, by writing the Results more concisely.

3. Change ‘N-metil-8-metoxiflindersin’ to ‘N-methyl-8-methoxyflindersin’ throughout the manuscript.

4. Change ‘metil’ to ‘methyl’, ‘metoxi’ to ‘methoxy’, and ‘hidroxi’ to ‘hydroxy’ throughout the manuscript.

5. Consequently use abbreviations such EC (or CE?), SI (or IS?), etc.

Specific comments

TITLE

1. Change to: ‘Antileishmanial activity of synthetic analogues of the quinoline alkaloid N-methyl-8-methoxyflindersin’’.

INTRODUCTION

1. Lines 54-55: Change to: ‘The first therapeutic choice in patients with CL is based on the intravenous or intralesional administration of pentavalent antimony, …’

2. Lines 57-59: Nonetheless, these medicaments could produce adverse effects associated with cardiotoxicity, hepatogenicity, nefrotoxicity or even teratogenic effects in the case of miltefosine (give references).

3. ‘Moreover, the prolonged treatment schemes and the parenteral administration way (intramuscular or intravenous), …’. Delete ‘way (intramuscular or intravenous)’.

4. Lines 66-68: Change to: ‘… the quinolinic alkaloids, which belong to the secondary metabolites and are mainly found in plants of the family Rutaceae, …’

5. Lines 69-71: Change to: ‘In addition, the 2-substituted quinolinic alkaloids chimanine D and B isolated from the stembark of the Rutacea Galipea longiflora K. Krause, showed activity against promastigotes of L. braziliensis and L. donovani (12, 13).

6. Lines 72-74: Change to: ‘In a previous study, the quinolinic alkaloid 7-methoxy -2,2- dimethyl-2H,5H,6H73 pyrano [3,2-c]quinolin-5-one or N-methyl-8-methoxyflindersine (1) was isolated from the leaves of the Rutaceae Raputia heptaphyla (14).

7. Lines 74-74: The compound had a direct effect against promastigotes of L. (V) panamensis and reduced the number of internalized parasites in dendritic cells (give references).

8. Lines 75-78: Change to: ‘However, the relatively low amounts of this compound obtained by extraction from its natural source (give references) and its complex structure prohibits its synthesis (give references). These disadvantages make it difficult to obtain sufficient material for preclinical studies to validate its therapeutic potential.

METHODS AND MATERIALS’

1. Change to ‘MATERIALS AND METHODS’

2. Make four subsections (and think about proper headings):

- In silico studies

- In vitro antileishmanial activity

- In vivo antileishmanial activity

- Biochemical and structural changes in infected macrophages

3. Line 240. Change ‘Statistical analysis’ to ‘Data processing and statistical analysis’

RESULTS

1. Too many subsections and the headings are too long.

2. The authors may consider subdividing this section according to the subdivision of the MATERIALS AND METHODS and think of headings based on those (but not identical to those) of the four subsections in the MATERIALS AND METHODS.

3. Lines 306-307. Change to: 'Cytotoxic and antileishmanial activity of quinolinic alkaloids compounds against 307 hMDM and L. (V) panamensis promastigotes and intracellular amastigotes.'

DISCUSSION

1. This section comprises about five pages which is too long to hold the attention of the reader. As mentioned before, too extensive wording distracts from the message the authors want to convey.

REFERENCES

1 The references in the reference list must consequently be given according to the format of the journal

6. PLOS authors have the option to publish the peer review history of their article (what does this mean?). If published, this will include your full peer review and any attached files.

Reviewer #1: **Yes: **Dennis R.A. Mans

---

## [Author Response · Author response to Decision Letter 0]

18 Sep 2020

- The references on this document have been management using the Mendeley software with the respective template of the journal. 

- English of this manuscript have been review by a native speaker.

- The “Discussion” has been revised, discussed and re-focused in order to make it more concisely.

- The changes proposed by the reviewer have made through all the document.

- “Materials and methods” section has been ordered according to the comment proposed by the reviewer.

- The “results” section has been ordered according to the sub-sections using for the presentation of the “Materials and methods”.

- Finally, due to one of the observations, the order of the presentation of the "materials and methods" and the "results" has change, reason by which the order of presentation of the different figures has change also. For this situation, the present version of the manuscript is accompanied by the figures with their new numeration.

More in detail, our point-by-point responses to the reviewers’ comments and the changes made are shown in table of the letter "Response to the reviewer".

---

## [Decision Letter · Decision Letter 1]

6 Oct 2020

PONE-D-20-17746R1

Antileishmanial activity of synthetic analogues of the quinoline alkaloid N-methyl-8-methoxyflindersin

PLOS ONE

Dear Dr. Delgado,

Thank you for submitting your manuscript to PLOS ONE. After careful consideration, we feel that it has merit but does not fully meet PLOS ONE’s publication criteria as it currently stands. Therefore, we invite you to submit a revised version of the manuscript that addresses the points raised during the review process.

You must address all issues raised by the expert reviewer. Also the use of English is still an issue. 

We look forward to receiving your revised manuscript.

Kind regards,

Henk D. F. H. Schallig, Ph.D

Academic Editor

PLOS ONE

Additional Editor Comments (if provided):

Please see comments of reviewer, these must be well addressed

Use of English is still an issue. Consult native speaking person or certified translator. provide proof that this has been done

Reviewers' comments:

Reviewer's Responses to Questions

**Comments to the Author**

1. If the authors have adequately addressed your comments raised in a previous round of review and you feel that this manuscript is now acceptable for publication, you may indicate that here to bypass the “Comments to the Author” section, enter your conflict of interest statement in the “Confidential to Editor” section, and submit your "Accept" recommendation.

Reviewer #1: (No Response)

2. Is the manuscript technically sound, and do the data support the conclusions?

Reviewer #1: Yes

3. Has the statistical analysis been performed appropriately and rigorously? 

Reviewer #1: Yes

4. Have the authors made all data underlying the findings in their manuscript fully available?

Reviewer #1: Yes

5. Is the manuscript presented in an intelligible fashion and written in standard English?

Reviewer #1: Yes

6. Review Comments to the Author

Reviewer #1: General comments

This manuscript has improved with respect to structure and readability when compared to the first version. However, even though the authors have a native English speaker review the manuscript before resubmitting it, there are still some linguistic issues that must taken care of. The authors may consider asking a native English speaker for a second opinion.

Specific comments

Title

Change title to: “Antileishmanial activity of synthetic analogues of the naturally occurring quinolone alkaloid N-methyl-8-methoxyflindersin”

Change short title to: “Potential antileishmanial activity of quinoline alkaloids”

Abstract

This section is okay, but the mistakes in lines 31-32 and 41-42 must be corrected

Indicate in lines 33-35 that these were in vitro studies.

Introduction

• Lines 56-59. Make two separate sentences. Thus: “The first therapeutic choice for CL patients is based on the intravenous or intralesional administration of pentavalent antimony [refs]. Oral administration of miltefosine or intravenous administration of amphotericin B have also been recommended as second therapeutic possibilities [5,6].”

• Lines 59-63. Make two separate sentences, improve the language, and clearer express the relationship between the statements in both sentences. Thus: “Nonetheless However, these drugs could produce adverse effects associated with such as cardiotoxicity, liver damage, nephrotoxicity, or even teratogenic effects in the case of miltefosine [7,8]. These drawbacks along with prolonged treatment schemes, and parenteral administration, can lead to non-compliance, and abandonment of prescribed treatment, and the consequent emergence of drug-resistant parasites could therefore occur [6,7,9].”

• Line 65. “….. and safer for patients (better adherence and less toxic effects) [10,11].” The authors must properly phrase this.

• Lines 66-69. The authors must properly introduce the quinoline alkaloids by giving some relevant background information about these compounds.

• Line 80. The authors should connect this alinea with the previous one by using, for instance, the expressions “Therefore, ……” or For this reason,”……..”.

Materials and Methods

• The authors may consider substituting the title of the first subsection by “Screening for synthetic analogues of N-methyl-8-methoxyflindersine”

• The authors may consider combining the “subsection “Isolating human monocyte-derived macrophages” with the subsection “Antileishmanial activity in L. (V.) panamensis intracellular amastigotes”

• The authors may consider placing the subsections “Inducing cell death in human monocyte-derived macrophages”, “Evaluating reactive oxygen species (ROS) and nitric oxide (NO) production in hMDM” (no abbreviations in section title), and “Transmission electron microscopy of L. (V) panamensis infected macrophages” immediately behind the subsection “Antileishmanial activity in L. (V.) panamensis intracellular amastigotes”

• The authors may consider placing the “Ethics statement” to the very end of the “Materials and Methods section

• The authors may consider placing the data processing of each methodology under the corresponding subsection and rewrite the subsection “Data processing and statistical analysis” to a subsection “Statistical analysis”.

Results

• The authors may consider thinking up better titles for the subsections. For instance, “In silico studies” may be substituted by “Synthetic analogues of N-methyl-8-methoxyflindersine”

• Similarly, the authors may come up with better titles than “In vitro cytotoxic and antileishmanial activity”, “Evaluating antileishmanial activity and safety in vivo”, etc.

Discussion

• The authors must properly connect the second alinea to the first. Just mentioning that “Quinoline alkaloids have been tested for their strong antiparasitic properties [13,14,33]”(line 449). For instance, the authors may take their time and mention that quinoline alkaloids including N-methyl-8-methoxyflindersine, have shown antileishmanial activity in vitro, and that synthetic analogues of N-methyl-8-methoxyflindersine have been tested in the current study. Etc., etc. The authirs should take their time explaining tjeri train of thiough, obviosuly, without exaggerating.

References

• The authors must make sure that all references in their reference list are according to the format of PLOS One. That is, with the journal names properly abbreviated.

7. PLOS authors have the option to publish the peer review history of their article (what does this mean?). If published, this will include your full peer review and any attached files.

Reviewer #1: **Yes: **Dennis R.A. Mans

---

## [Author Response · Author response to Decision Letter 1]

18 Nov 2020

Comment reviewer 1 (CR1): This manuscript has improved with respect to structure and readability when compared to the first version. However, even though the authors have a native English speaker review the manuscript before resubmitting it, there are still some linguistic issues that must taken care of. The authors may consider asking a native English speaker for a second opinion.

Response:The English of the article has been reviewed by a new official translator native to the language (Enago).

CR2: Change title to: “Antileishmanial activity of synthetic analogues of the naturally occurring quinolone alkaloid N-methyl-8-methoxyflindersin”

Response: The title was changed to "Antileishmanial activity of synthetic analogs of the naturally occurring quinolone alkaloid N-methyl-8-methoxyflindersin"

CR3: Change short title to: “Potential antileishmanial activity of quinoline alkaloids”

Response: The short title was changed to "Quinoline alkaloid and its potential antileishmanial activity" to "Potential antileishmanial activity of quinoline alkaloids".

CR4: About abstract and introduction mistakes.

Response: Mistakes in typing and references was corrected. 

CR5: The authors may consider substituting the title of the first subsection by “Screening for synthetic analogues of N-methyl-8-methoxyflindersine”.

Response: The change proposed has made, therefore the title changes from "Screening for synthetic compounds analogous to the natural quinoline alkaloid" to "Screening for synthetic analogs of compound 1”.

CR6: The authors may consider combining the “subsection “Isolating human monocyte-derived macrophages” with the subsection “Antileishmanial activity in L. (V.) panamensis intracellular amastigotes”.

Response: It is considered not to make changes in the order of the sub-sections, to make it easier to read separately, the requirements of cell culture and each technical apart; because they all share the same cellular model (hMDMs). But we change the title to “Isolating hMDMs”.

CR7: The authors may consider placing the “Ethics statement” to the very end of the “Materials and Methods section.

Response: The subsection was moved by the end of the Methods section.

CR8: The authors may consider placing the data processing of each methodology under the corresponding subsection and rewrite the subsection “Data processing and statistical analysis” to a subsection “Statistical analysis”.

Response: The title of the subsection changed from "Data processing and statistical analysis" to subsection "Statistical analysis".

CR9: The authors may consider thinking up better titles for the subsections. For instance, “In silico studies” may be substituted by “Synthetic analogues of N-methyl-8-methoxyflindersine”.

Response: Replaced with "Synthetic analogues of compound 1".

CR10: Similarly, the authors may come up with better titles than “In vitro cytotoxic and antileishmanial activity”, “Evaluating antileishmanial activity and safety in vivo”, etc.

Response: The changes suggested has made. The titles were replaced: “In vitro evaluation of antileishmanial activity”, “Biochemical (Nitric Oxide and Reactive Oxygen Species production) and structural changes in infected hMDMs”, “Induction of cell apoptosis in L. (V.) panamensis- infected hMDMs”, “L. (V.) panamensis-infected hMDM ultrastructural alterations”, “Ultrastructural modifications of quinoline alkaloid-like compounds in L. (V.) panamensis intracellular amastigotes”.

CR11: The authors must properly connect the second alinea to the first. Just mentioning that “Quinoline alkaloids have been tested for their strong antiparasitic properties [13,14,33]”(line 449). For instance, the authors may take their time and mention that quinoline alkaloids including N-methyl-8-methoxyflindersine, have shown antileishmanial activity in vitro, and that synthetic analogues of N-methyl-8-methoxyflindersine have been tested in the current study. Etc., etc. The authors should take their time explaining tjeri train of though, obviosuly, without exaggerating.

Response: The recommendation was implemented and the emphasis has made on the relevance of the compound N-methyl-8-methyl flindersine as antileishmanial, and the In vitro evaluation of synthetic compounds.

---

## [Editor Report · Decision Letter 2]

20 Nov 2020

Antileishmanial activity of synthetic analogs of the naturally occurring quinolone alkaloid N-methyl-8-methoxyflindersin

PONE-D-20-17746R2

Dear Dr. Delgado,

We’re pleased to inform you that your manuscript has been judged scientifically suitable for publication and will be formally accepted for publication once it meets all outstanding technical requirements.

Kind regards,

Henk D. F. H. Schallig, Ph.D

Academic Editor

PLOS ONE
---

## [Editor Report · Acceptance letter]

26 Nov 2020

PONE-D-20-17746R2 

Antileishmanial activity of synthetic analogs of the naturally occurring quinolone alkaloid N-methyl-8-methoxyflindersin 

Dear Dr. Delgado:

I'm pleased to inform you that your manuscript has been deemed suitable for publication in PLOS ONE. Congratulations! Your manuscript is now with our production department. 

Kind regards, 

on behalf of

Dr. Henk D. F. H. Schallig 

Academic Editor

PLOS ONE